# Arginine Homopeptide of 11 Residues as a Model of Cell-Penetrating Peptides in the Interaction with Bacterial Membranes

**DOI:** 10.3390/membranes12121180

**Published:** 2022-11-24

**Authors:** Mónica Aróstica, Roberto Rojas, Luis Felipe Aguilar, Patricio Carvajal-Rondanelli, Fernando Albericio, Fanny Guzmán, Constanza Cárdenas

**Affiliations:** 1Doctorado en Biotecnología, Pontificia Universidad Católica de Valparaíso y Universidad Técnica Federico Santa María, Valparaíso 2373223, Chile; 2Facultad de Medicina Veterinaria y Agronomía, Universidad de Las Américas, Sede Los Castaños, 7 Norte 1348, Viña del Mar 2531098, Chile; 3Instituto de Química, Pontificia Universidad Católica de Valparaíso, Av. Universidad 330, Valparaíso 2373223, Chile; 4Escuela de Alimentos, Facultad de Ciencias Agronómicas y de los Alimentos, Pontificia Universidad Católica de Valparaíso, Waddington 716, Valparaíso 2360100, Chile; 5Department of Organic Chemistry and CIBER-BBN, Networking Centre on Bioengineering, Biomaterials and Nanomedicine, University of Barcelona, Martí i Franqués, 1, 08028 Barcelona, Spain; 6School of Chemistry, University of KwaZulu-Natal, King Edward Avenue, Scottsville, Durban 4001, South Africa; 7Núcleo de Biotecnología Curauma, Pontificia Universidad Católica de Valparaíso, Av. Universidad 330, Valparaíso 2373223, Chile

**Keywords:** arginine homopeptide, secondary structure PPII, circular dichroism, model membrane interactions, antibacterial assays

## Abstract

Cell-penetrating peptides rich in arginine are good candidates to be considered as antibacterial compounds, since peptides have a lower chance of generating resistance than commonly used antibiotics. Model homopeptides are a useful tool in the study of activity and its correlation with a secondary structure, constituting an initial step in the construction of functional heteropeptides. In this report, the 11-residue arginine homopeptide (R11) was used to determine its antimicrobial activity against *Staphylococcus aureus* and *Escherichia coli* and the effect on the secondary structure, caused by the substitution of the arginine residue by the amino acids Ala, Pro, Leu and Trp, using the scanning technique. As a result, most of the substitutions improved the antibacterial activity, and nine peptides were significantly more active than R11 against the two tested bacteria. The cell-penetrating characteristic of the peptides was verified by SYTOX green assay, with no disruption to the bacterial membranes. Regarding the secondary structure in four different media—PBS, TFE, *E. coli* membrane extracts and DMPG vesicles—the polyproline II structure, the one of the parent R11, was not altered by unique substitutions, although the secondary structure of the peptides was best defined in *E. coli* membrane extract. This work aimed to shed light on the behavior of the interaction model of penetrating peptides and bacterial membranes to enhance the development of functional heteropeptides.

## 1. Introduction

Antimicrobial peptides (AMPs) are considered as alternative compounds to antibiotics, thanks to their multiple action pathways that make them less likely to generate resistance [1,2,3,4,5]. AMPs can be classified according to different criteria, which link with their amino acid composition, their secondary structure, their activity or their source [6]. Regarding their mechanism of action, they can be classified in two large groups: AMPs whose target is the cell membrane and AMPs who have intracellular targets. The first group corresponds to the AMPs that have been best studied, and at least three models of action of these peptides on the cell membrane have been proposed: (1) Pore formation by toroidal model; (2) Pore formation by barrel-stave model and (3) Carpet model, where the accumulation of peptide on the membrane produces a detergent-like effect. The second group includes various mechanisms of action, such as the binding to nucleic acids to interfere with replication, RNA synthesis and protein synthesis, among others [2,7]. AMPs usually exert their action through combined action mechanisms, thus avoiding the generation of resistance. Within the great diversity of AMPS, cationic peptides are perhaps best described with the plasma membrane as their main target through the generation of pores.

A particular class of cationic peptides comprises the arginine-rich peptides, which are included within a class of peptides known as cell-penetrating peptides (CPP), since they exert their activity against intracellular targets and do not generate pores or membrane disruption and are able to penetrate the membrane regardless of receptors [8].

CPPs peptides can cross the membrane through two independent mechanisms: one dependent on endocytosis and the other by independent translocation. However, the mechanisms of action of CPP have not been completely elucidated [9,10].

Model homopeptides are a useful tool for studying activity and its correlation with a secondary structure, and they also constitute an initial step in the construction of functional heteropeptides. Arginine has been described as an important amino acid in the activity of CPP [9].

Arginine homopeptides form a polyproline II helix-like secondary structure. It has also been seen that these homopeptides have activity against a broad spectrum of bacteria [11]. 

In previous pieces of work, it was determined that arginine homopeptides between 7–14 residues presented activity against several bacterial strains, with the 11-residue homopeptide being the one that presented the highest antimicrobial activity and lowest toxicity [11,12]. In the present work, the 11-residue arginine homopeptide (shortened to R11 for ease hereafter) is used to determine the effect on the antimicrobial activity and, on the secondary structure, is caused by the substitution of arginine in each position by four different amino acids, Ala, Pro, Leu and Trp, by using the scanning technique [13].

The four amino acids used for scanning display various physicochemical characteristics, such as size, hydrophobicity and flexibility, also being amino acids that have been reported in peptides with antibacterial activity. Ala is a neutral amino acid that is the most common choice used for screening, which would be equivalent to removing the side chain. Pro is known as a secondary-structure-breaking residue [14] because its side chain binds to the alpha nitrogen (imino acid), causing a twist in the backbone. Leu was chosen as the hydrophobic amino acid with the largest side chain and Trp as the aromatic hydrophobic residue with the largest side chain, also reported as important for the interactions of AMPs and CPPs [15]. 

45 peptides were chemically synthesized by solid phase and their antimicrobial activity was studied on *Escherichia coli* and *Staphylococcus aureus*.

Their secondary structure was characterized by circular dichroism (CD) in four different media: 30% trifluoroethanol (TFE) that is a common media used for peptide CD spectra due to their capacity for lowering the dielectric constant and favoring intramolecular interactions [16]; phosphate-buffered saline (PBS), which is the common medium used in biological experimental assays; and two models of vesicles, namely, permeable vesicles of 1,2-dimyristoyl-snglycero-3-phospoglycerol lipid (DMPG) and *Escherichia coli* membrane extract. DMPG is an anionic saturated phospholipid and is the major component in the membranes of Gram-positive bacteria, being widely used as a simple model to mimic bacterial membranes in order to study the interaction with cationic peptides [17]. *Escherichia coli* membrane extracts have a heterogeneous lipid composition. The main components of the inner membrane are zwitterionic phosphatidylethanolamine (PE), cardiolipin (CL) and phosphatidylglycerol (PG) [18].

Herein, the impact of the substitution of these amino acids into both activity and secondary structure is analyzed.

## 2. Materials and Methods

### 2.1. Peptide Synthesis and Purification

Simultaneous synthesis of 45 peptides, R11, 11 peptides for each scanning (Ala, Pro, Leu and Trp), and the 13 residues peptide derived from *Bothrops asper* snake venom (KKWRWWLKALAKK) [19] used as positive control peptide for SYTOX green assay, was performed by the “tea bag” technique [15,16] using the Fmoc/tBu (9-fluorenylmethoxycarbonyl) solid phase synthesis strategy. Briefly, Fmoc Rink-amide resin with 0.59 mmol/g loading was used (Iris Biotech GmbH, Marktredwitz, Germany). N-[(1H-benzotriazol-1-yl)(dimethylamine)-methylene]-N-methylmethanaminium hexafluorophosphate N-oxide (HBTU, Iris Biotech GmbH) and N-[(1H-benzotriazol-1-yl)(dimethylamine)-methylene]-N-methylmethanaminium tetrafluoroborate N-oxide (TBTU, Iris Biotech GmbH) were used as activators. Oxyme pure (Sigma-Aldrich, St. Louis, MO, USA) was used as an additive and N,N-diisopropilethylamine (DIPEA, Merck) for “in situ” neutralization. DMF was used as main solvent (Merck). Cleavage from the resin was produced by using a mix of trifluoroacetic acid (TFA)/triisopropyl silane (TIS)/Milli-Q water (95/2.5/2.5 *v*/*v*). Peptides were characterized by RP-HPLC in a Water Corp Xbridge^tm^ BEH C18 column (100 × 4.6 mm 4.6 mm, 3.5 µm) and by mass spectrometry in UFLC-ESI Shimadzu LCMS-2020. Purification was performed with C18 cartridges (United Chemical Technologies, Bristol, PA, USA). Purified peptides were stored as lyophilized dry powders at −80 °C and were dissolved just before use.

### 2.2. Vesicles Preparation

Large unilamellar vesicles (LUV) were prepared using the extrusion method, as described in a previous piece of work [17]. Two lipid models were used: (a) *Escherichia coli* bacterial membrane extract and (b) 1,2-dimyristoyl-snglycero-3-phosphoglycerol lipid (DMPG) (Sigma-Aldrich). 

For the first case, it was necessary to obtain the natural lipids of the bacterial membranes, for which the bacteria were grown to an OD 600 nm of 2.0. Then, the bacterial suspension was sonicated (3 times at 100 watts) and centrifuged at 30,000 rpm for 10 min at 4 °C; finally, the bacteria sediment was resuspended in chloroform.

The two types of lipids dissolved in CHCl_3_ (400 µM) were dried in glass tubes under a gaseous nitrogen flow. The lipid film was resuspended in 15 mL of MilliQ water and left in a water bath for 5 min at 50 °C.

The vesicles were prepared using a cycle of freezing under liquid nitrogen and thawing at 50 °C 10 times and a subsequent extrusion through a 0.4 µm polycarbonate membrane (Millipore, Tullagreen, Ireland) to guarantee the homogeneity of the vesicles [11,18,19].

### 2.3. Antibacterial Assays

The strains used, *Staphylococcus aureus* (ATCC 25923) and *Escherichia coli* (ML35), were obtained from the Public Health Institute of Chile (ISP).

To determine the antibacterial activity of the peptides, a microplate dilution test was performed. The necessary calculations were completed with a bacterial concentration of 1 × 10^7^ CFU/mL and the cells were exposed to 50 µL of Tryptic soy broth (TSB) with 1%, 10 mM of Hepes buffer and 10, 20 and 30 μM of peptide for 1 h at 37 °C. Finally, a serial dilution was performed and incubated for 18 h at 37 °C in complete TSB, and the minimum bactericidal concentration (MCB) was determined depending on the turbidity of the plate [20,21]. All experiments were completed in duplicate. A synthetic peptide of sequence KKWRWWLKALAKK from the venom of the *Bothrops asper* snake was used as a positive control for growth inhibition [11,22]. 

### 2.4. Membrane Permeabilization by SYTOX Green Assay

To verify the effect of peptides on cell membrane integrity, a SYTOX green uptake assay was used, as previously described. Briefly, *E coli* and *S. aureus* cultures in exponential phase of growth were centrifuged at 2000× *g* for 5 min and washed with PBS 1X to prepare a solution at a concentration of 1 × 10^6^ CFU/mL in PBS 1X. Then, in 100 μL qPCR tubes, 90 µL of the bacterial solution, 5 µL of peptide at 200 µM and 5 µL of SYTOX Green dye at 100 µM were added. The q-PCR tubes were placed in an Applied Biosystems StepOnePlus™ Real-Time PCR Systems thermocycler, programmed with 30 cycles for one minute at 30 °C, using the Syber green filter, recording the fluorescence data at the end of each cycle. Each assay was performed in triplicate. 

### 2.5. Hemolysis Assay

The hemolytic activity of the peptides was obtained, according to a previously described procedure [22]. Briefly, blood samples obtained from a healthy individual were centrifuged at 2000 *g* for 10 min at 4 °C and then washed with PBS 1X three times. The supernatant was diluted with PBS 1X until reaching a concentration of 6 × 10^8^ cells/mL. Assays were performed in duplicate with the peptides at concentrations of 5 and 50 µM, including AMP from Bothrops asper. Triton X-100 (Sigma Aldrich) at 0.5% was used as positive control and PBS 1X as negative control. The assay is performed by mixing 65 µL of the cell suspension with 65 µL of each peptide at the indicated concentrations, and 65 µL of each control, incubating at 37 °C for 1 h. The samples are centrifuged at 3000 *g* for 5 min; 80 µL of supernatant is taken and placed in a 96-well plate together with 80 µL of milliQ water to finally measure its absorbance at 540 nm.

Hemolysis is calculated according to Equation (1).
(1)%Hemolysis=((ASA100%)−A0%)×100

### 2.6. Circular Dichroism Analysis

The secondary structure of the synthesized peptides was determined by CD with a Jasco J-815 CD Spectrometer coupled to a Peltier Jasco CDF-426 S/15 temperature controller (Jasco Corp., Tokyo, Japan). The measurements were obtained in the distant ultraviolet range (190–250 nm), using quartz cells with 0.1 cm path length and 1 nm bandwidth. Each spectrum was measured three times in continuous scanning mode with 100 nm/min scanning speed and to response time of 2 s. The data analysis was completed using Spectra Manager Software version 2.0.

The four media used were: 2 mM PBS prepared from tablets, according to the suppliers’ instructions (Sigma Aldrich, Merck, Darmstadt, Germany); 30% 2,2,2-trifluorethanol (TFE, Merck, Darmstadt, Germany) in water; DMPG LUV with 400 µM phospholipid concentration of (Sigma Aldrich, Merck, Darmstadt, Germany); and LUV of *E. coli* membrane extract with 66 µM phospholipid concentration. Peptide concentration for CD measures was of 0.1 mM. The working temperature was 37 °C [12,22].

### 2.7. Statistical Analysis

For SYTOX green permeabilization assay, the results were analyzed with GraphPad Prism 8.0.2 (La Jolla, CA, USA, www.graphpad.com) with two way ANOVA and Tukey’s multiple comparison test.

## 3. Results

### 3.1. Peptide Synthesis and Peptides Chracterization

A total of 45 peptides were synthesized, purified, and characterized by RP-HPLC and mass spectrometry; Appendix A present a summary of the sequences and main features of the synthesized peptides. 

The molecular mass determined for the synthesized peptides coincides with the calculated mass, and the RP-HPLC results did not show significant changes in retention times with respect to the homopeptide R11. This is a first indication that the substitution of one single amino acid residue does not produce a significant alteration of the hydrophilicity and/or conformation of the peptides with respect to the parent R11.

### 3.2. Antibacterial Activity

Results of the microplate tests for the four scannings are summarized in Table 1, expressed as the minimum bactericidal concentration (MBC) of each peptide in μM.

The antibacterial activity against the two bacteria was generally increased by the substitutions made on the homopeptide, with the exception of P10, L6, W1 and W2 for *S. aureus* and A1, A9, P4, L6, W2 and W3 for *E. coli* (Table 1 and Figure 1).

Ala scanning increased the activity against *S. aureus* in all cases; the effect against *E. coli* was less noticeable, with the replacements in A3 and A10 standing out as the most active against the two bacteria. 

Pro scanning showed increased activity against *S. aureus* in almost all cases, with a tendency to be less significant towards the C-terminus of the sequence; replacements towards the center of the peptide chain negatively affect the activity against *E. coli*. The peptides with increased activity against the two bacteria were P1, P6 and P7.

In the case of Leu scanning, the replacements in the center of the chain decreased the activity with respect to R11, for both bacteria, and the peptides that presented increased activity against them were L2, L3 and L9.

Trp scanning was the one showing the lowest increase in activity with respect to R11. Changes towards the N-terminus of the peptide cause decreased activity and the replacement at W7 produced a peptide with increased activity against both bacteria.

### 3.3. Sytox Green Permeabilization Assay

Peptides with the best results in terms of MBC (indicated by blue boxes in Table 1 and asterisks in Figure 1) and the antimicrobial peptide from Bothrops asper venom [11,22] as a positive control were used in the SYTOX green assay with the two bacteria. The fluorescence emitted by the dye is proportional to the cells whose membrane has been disrupted. In this assay, the peptides behaved in a similar way to a previous piece of work [11] and, as can be seen in Figure 2, only the positive control showed high fluorescence. Peptide R11 and its analogs showed low fluorescence values similar to the negative control, indicating that their effect is not on the membrane itself and that there is no membrane disruption, confirming their potential attributes as cell-penetrating peptides.

Statistical analysis showed significant differences in all peptides both in comparison to the control and between each other. A summary of these results can be seen in Appendix A. 

### 3.4. Hemolytic Activity

Peptides were tested at concentrations of the lowest MBC, 5 µM, and 10 times this concentration at 50 µM. None of the peptides exhibited hemolytic activity; hemolysis was only observed with the positive control, 0.15% triton X-100 (Appendix A).

### 3.5. Peptides Secondary Structure in Different Media

Previous studies showed that R11 tends to form a polyproline II type helix (PPII) that stabilizes at low temperatures [11], which is characterized by a weak positive band at 218 nm and a strong negative band at 195 nm. In this study, the temperature used for the determination of secondary structure in the four media studied was 37 °C, simulating the physiological temperature.

Secondary structure was assessed by two methods. The first one used CDPro analysis with the reference database SP37A and CONTIN method [23], which includes five structural classes: alfa helix (H), beta strand (S), beta turn (T), polyproline II (PPII) and unordered structure (U); the CDPro-CONTIN [24,25] is implemented in a spectra analysis of the J-815 CD spectrometer. The second method is the calculation of the PPII content, according to Equation (2) [26]. Due to the displacements and distortions of the spectra with the different amino acid scans, the wavelength at which the maximum is located can vary; for this reason, the wavelength selected for the calculation was 218 nm.
(2)$$\%\mathrm{PPII}=\frac{{\left[\theta \right]}_{218}+6100}{13700}\times 100$$
with [θ]_218_ being the ellipticity at 218 nm.

The behavior of peptides, in relation to their secondary structure, is dependent on the medium in which it is determined (Table 2 and Appendix A). In this work, two aqueous media, 2 mM PBS and 30% TFE, and two membrane models, *E. coli* membrane extract and DMPG vesicles, were used. 

In the case of the calculation of PPII through Equation (2), almost all values are above 40%; regarding R11, there are variations depending on the amino acid used in the scan. It should be noted that, in addition to the analysis, it is important to consider the shape of the curves. According to the CDPro analysis (Appendix A), the structural class with the highest representation is the unordered structure (Unrd), and there are no appreciable changes between R11 and the different scannings, with some exceptions, such as the replacement of L3 in PBS and *E. coli* membranes, as well as W7 in DMPG.

Results for each amino acid scanning are detailed below.

#### 3.5.1. Ala Scanning

Ala appears to be the amino acid that exerts the least perturbations on the secondary structure. The CD spectra have a PPII trend, and “in-phase” minima and maxima are observed (Figure 3). The % PPII is similar to the one in R11, being slightly higher in *E. coli* membranes (Appendix A). In regard to the shape of the curves, although the minimum and maximum values vary, they look well behaved, except for A2, A4 and A6, that tend to flatten in almost all media, especially in PBS, where they lose the minimum along with A8 and A11 (Figure 1).

#### 3.5.2. Pro Scanning

The replacements by Pro generated a distortion in the CD spectra (Figure 4), making the curves look uneven and also presenting displacements in the minimum and maximum positions. The content of PPII relative to R11 is higher only in some cases, such as P8 in TFE and N-terminal replacements (P1, P2 and P3) and P10 in *E. coli*. However, as mentioned above, the shape of the curve must also be considered, and, in this case, the effect of Pro as a secondary structure breaker is noted.

#### 3.5.3. Leu Scanning

In the case of Leu replacements, the range of ellipticity values was greatly extended, both in the minimum and maximum values (Figure 5). The best-behaved curves are seen in TFE, although they are not observed “in phase”, as in the case of Ala. In PBS, the curves are quite distorted, with the exception of L1 and L10. In DMPG and *E. coli* membrane extracts, the perturbations occur at shorter wavelengths towards the position of the minimum. In this case, due to the effect of the ellipticity scale, the PPII content is greater than for R11 in all cases (Appendix A).

#### 3.5.4. Trp Scanning

Similarly to Leu, in the replacements by Trp the ellipticity scale widens significantly, and the best performing curves are seen in TFE (Figure 6). In PBS and DMPG, the only curve that maintains the trend is R11, with large variations in the minimum at short wavelengths. In *E. coli*, although they still look very rippled, the central replacements tend to show better-behaved curves (W6, W7, W8, W9 and W10, Figure 6). In 30% TFE, an increase in the %PPII was observed with respect to R11, with almost all the substitutions. In DMPG, the highest content of PPII was observed in R11, and in PBS and *E. coli* only some had a %PPII higher than R11 (Appendix A).

## 4. Discussion

Arg-rich peptides have been studied as cell-penetrating peptides and used as vehicles to internalize other molecules in eukaryotic cells and as antibacterial peptides with targets within the cell [27,28]. In this context, the interaction with the membrane has been the subject of various studies, due to its internalization capacity without causing membrane disruption.

It is well demonstrated that Arg-rich membrane-penetrating peptides enter the cell through a mechanism mediated by interaction with the negative groups of membrane lipids. In this mechanism, the exchange of counterions promotes the formation of amphiphilic lipid–amino acid complexes, helping the cell internalization process [27,28,29,30]. In this context, some amino acids seem to favor interaction with the membrane, as is the case of Trp, which has been reported to enhance the interaction of Arg-rich CPPs with membranes due to π–ion pair interactions, also depending on the number of Trp residues present in the sequence, and has been used as a modifier in Arg-rich peptides to increase their internalization capacity [30,31]. Leu has a similar effect because of its hydrophobic characteristics and Leu-Arg motifs that can translocate across lipid membranes with a residue position dependence [32]. In our case, the effect of single replacements of these hydrophobic amino acids in R11 does not seem to produce major changes in ionic interactions with membrane lipids, nor on the secondary structure of PPII. 

Ala replacements can be considered as neutral in terms of secondary structure, and the Pro scanning shows some distortion in the CD curves, as expected considering Pro is a structure breaker [33].

Regarding the antibacterial activity of Arg-rich peptides, it is known that their target is intracellular. One proposed mechanism is the binding of peptides to DNA, favored by the interaction of the guanidinium group of Arg with phosphate, blocking DNA polymerase and bacterial proliferation [34]. Its action on the second messenger c-di-GMP has also been suggested, preventing the formation of biofilm for *P. aeruginosa* [35]. The translocation capacity is also a factor that influences the antibacterial activity and would be related to the effective concentration of the peptide to exert its activity.

Some studies have shown that the helical secondary structure is important for the internalization of cell-penetrating peptides. The helical structure allows the side chains of the cationic amino acids to be aligned and this appears to be an important feature for the translocation of the peptides. Studies on polyproline II helical structures have shown that aligned guanidino groups exhibit more efficient translocation. The alignment of the charged groups and the amphipathicity of the peptides play an important role in the translocation of penetrating peptides [36].

In this work, unlike the secondary structure, the antibacterial activity was affected by the change of one amino acid in the original peptide R11. *S. aureus* was more affected than *E. coli* by both the original peptide and the analogues of the scannings. Although it is not possible to establish a pattern associated with the replacements, there are some cases to note: Leu in the center of the chain decreased the antibacterial activity, as did Trp in the N-terminal region of the peptide. In general, most of the replacements increased their activity in terms of MBC with respect to R11, and at least nine peptides with improved activity against the two bacteria were found (Table 1). 

The secondary structure of peptides determined by CD clearly depends on the medium in which it is measured. The analyses carried out with CDPro show the limitations of the algorithms for the calculation of the secondary structure, even when the database that includes the PPII within the structural classes has been used, and although it is limited and does not correlate with the spectra obtained, the behavior is similar for all the scannings. PBS, as the aqueous medium with higher dielectric constant, and DMPG, the anionic membrane model, showed the highest variations with the replacements of the hydrophobic amino acids, Leu and Trp, which agrees with the interaction exerted by these amino acids with the medium. Additionally, in CD spectra analysis it is also important to consider the shape of the curve, as can be seen in the case of Trp, where the presence of aromatic residues generated distortion of the curves due to the *n* → π* transitions, as reported [37,38].

## 5. Conclusions

The unique substitutions on the arginine homopeptide of 11 residues by four amino acids with different characteristics—Ala as a neutral amino acid, Pro as a structure breaker, Leu as a hydrophobic amino acid and Trp as an aromatic amino acid—showed an increase in their antibacterial activity. At least six analogues were obtained with higher activity than the parent R11 peptide against the two bacteria tested, although the substitutions did not produce a defined pattern of behavior.

In terms of secondary structure, no notable changes due to the substitutions were observed, but it should be noted that the type II polyproline helix tends to be better defined in interaction with *E. coli* membrane extracts for all peptides.

It would be necessary to carry out further experiments to establish in more detail the effect of each type of amino acid on the activity of R11 as a model peptide. The replacement by more than one residue would allow us to observe if there are changes in the secondary structure that promote a more efficient translocation. This may be the mechanism involved in enhancing the antibacterial activity by increasing the effective concentration within the cell. 

Homopeptides have been shown to be excellent templates for understanding the role of different amino acids in the secondary structure and biological activity of parent peptides. In this context, double positional scannings will be performed to delineate peptides de novo with the aim of developing new families of antimicrobial peptides.

## Figures and Tables

**Figure 1 membranes-12-01180-f001:**
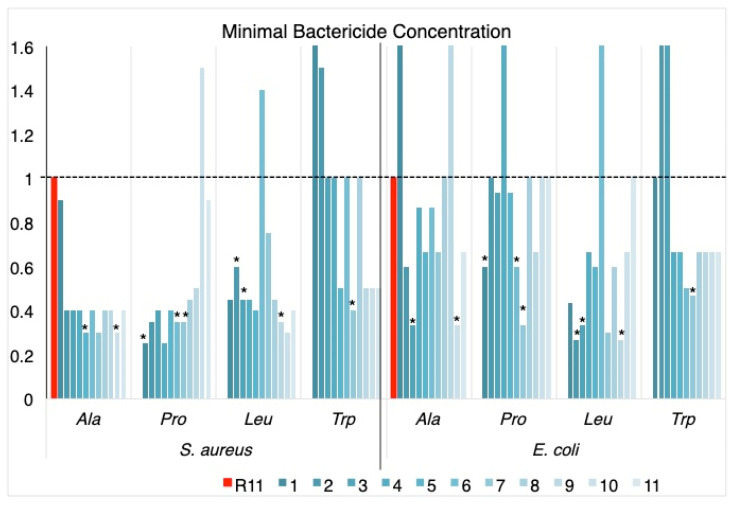
MBC values normalized with respect to R11 (MBC peptide/MBC R11). Replacements are shown in color gradient, from dark to light blue for positions 1 to 11, for each amino acid scanning. Black line denotes R11 value (red bars); asterisks (*) indicate the selected peptides with the lowest MBC against the two bacteria (Table 1).

**Figure 2 membranes-12-01180-f002:**
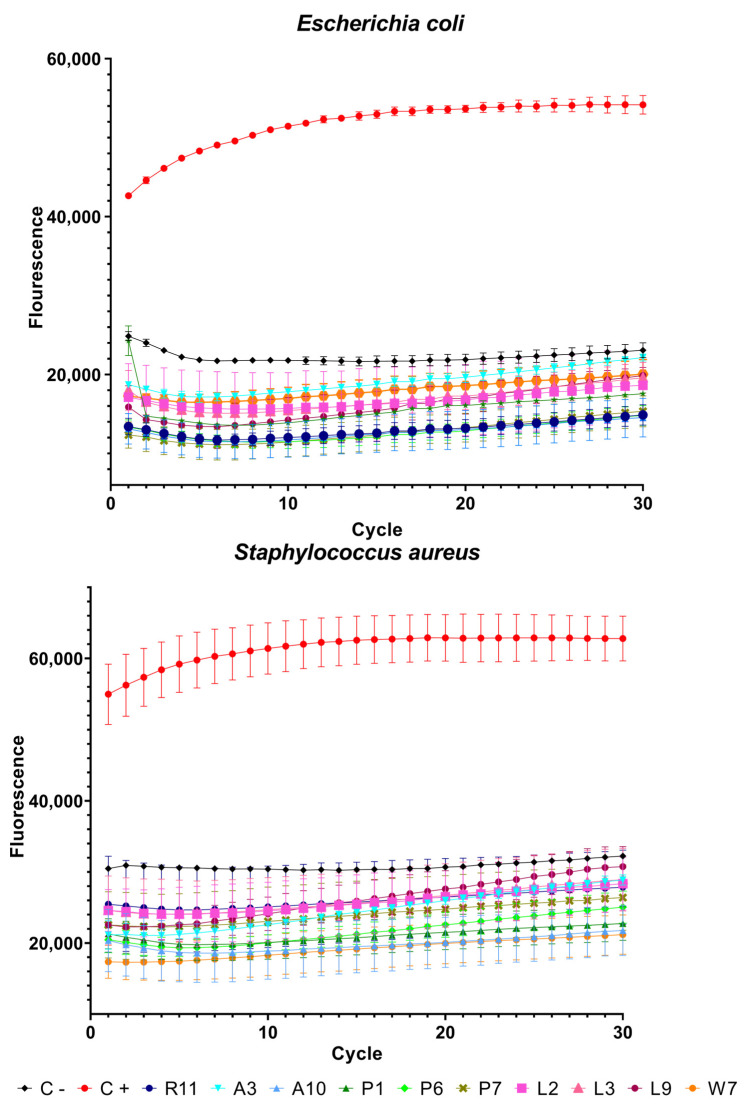
SYTOX green permeabilization assay. High fluorescence values indicate membrane permeabilization. Synthetic peptide KKWRWWLKALAKK was used as positive control.

**Figure 3 membranes-12-01180-f003:**
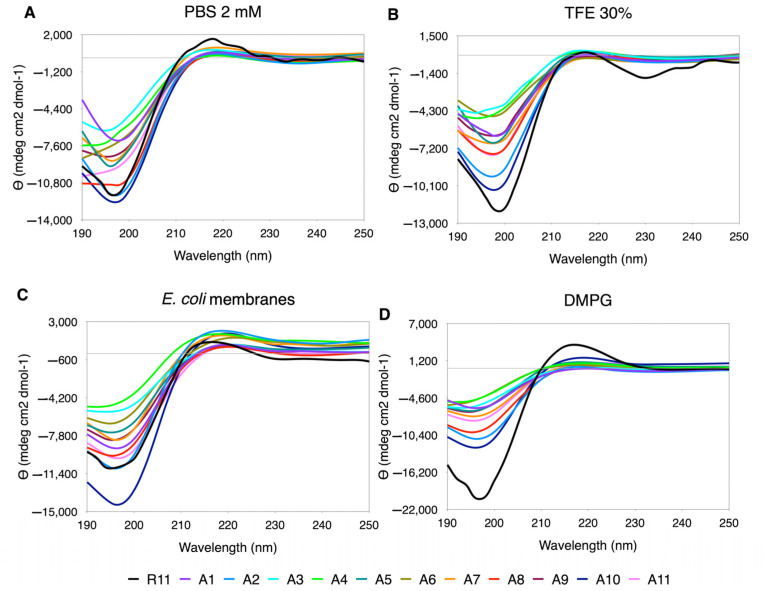
CD spectra for Ala scanning of R11 in the four media used. (**A**) 30% TFE. (**B**) 2 mM PBS. (**C**) DMPG. (**D**) *E. coli* membranes extract.

**Figure 4 membranes-12-01180-f004:**
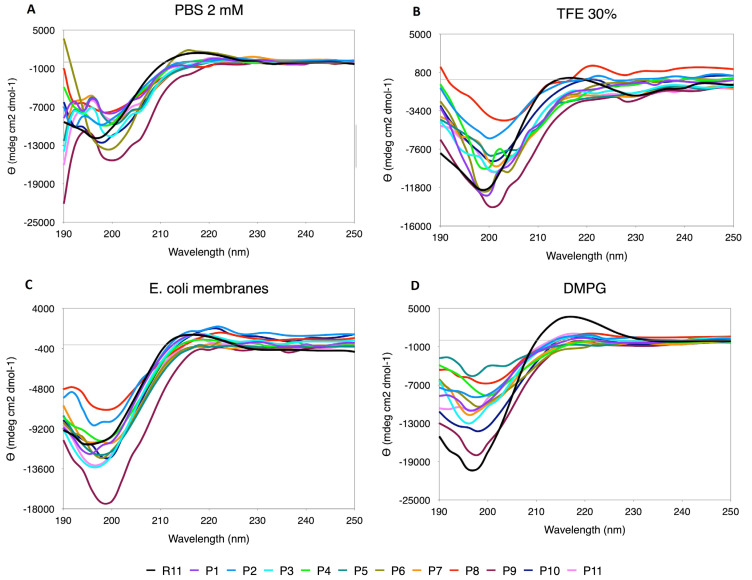
CD spectra for Pro scanning of R11 in the four media used. (**A**) TFE 30%. (**B**) PBS 2 mM. (**C**) DMPG. (**D**) *E. coli* membranes extract.

**Figure 5 membranes-12-01180-f005:**
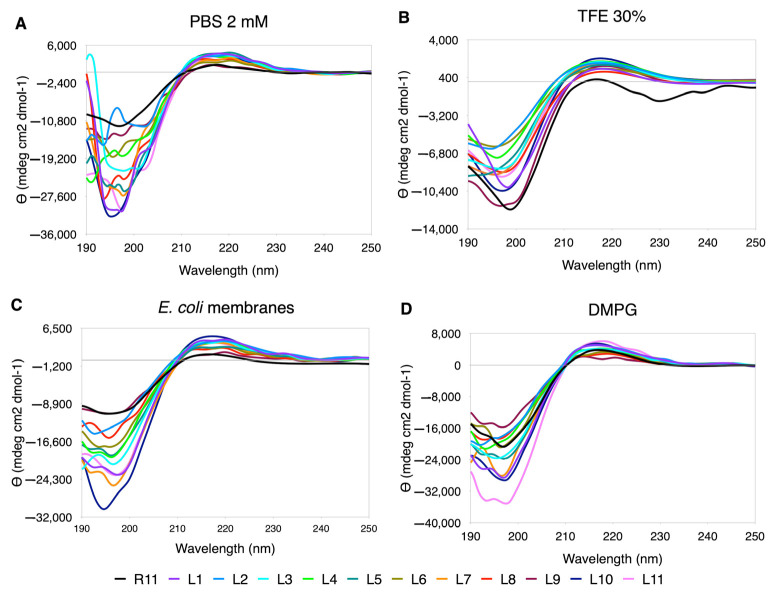
CD spectra for Leu scanning of R11 in the four media used. (**A**) TFE 30%. (**B**) PBS 2 mM. (**C**) DMPG. (**D**) *E. coli* membranes extract.

**Figure 6 membranes-12-01180-f006:**
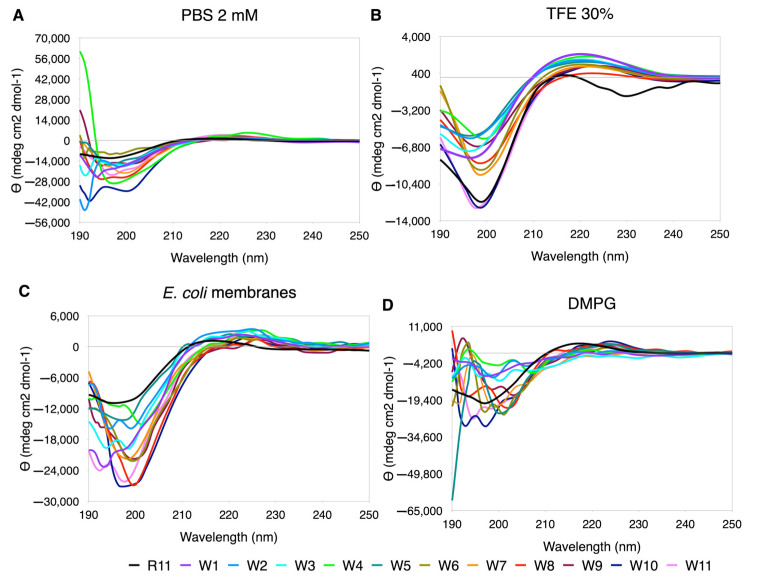
CD spectra for Trp scanning of R11 in the four media used. (**A**) TFE 30%. (**B**) PBS 2 mM. (**C**) DMPG. (**D**) *E. coli* membranes extract.

**Table 1 membranes-12-01180-t001:** Antibacterial activity against *S. aureus* and *E. coli* for R11 scanning. MBC is expressed in μM. The results are shaded in gray with the lowest values in light and the highest in dark. Blue boxes indicate peptides with lowest MBC for both bacteria.

Bacteria	Minimum Bactericidal Concentration (μM)											
Ala Scanning												
	**R11**	**A1**	**A2**	**A3**	**A4**	**A5**	**A6**	**A7**	**A8**	**A9**	**A10**	**A11**
*S. aureus*	20	18	8	8	8	6	8	6	8	8	6	8
*E. coli*	30	>30	18	10	26	20	26	20	30	>30	10	20
Pro Scanning												
	**R11**	**P1**	**P2**	**P3**	**P4**	**P5**	**P6**	**P7**	**P8**	**P9**	**P10**	**P11**
*S. aureus*	20	5	7	8	5	8	7	7	9	10	30	18
*E. coli*	30	18	30	28	>30	28	18	10	30	20	30	30
Leu Scanning												
	**R11**	**L1**	**L2**	**L3**	**L4**	**L5**	**L6**	**L7**	**L8**	**L9**	**L10**	**L11**
*S. aureus*	20	9	12	9	9	8	28	15	9	7	6	8
*E. coli*	30	13	8	10	20	18	>30	9	18	8	20	30
Trp Scanning												
	**R11**	**W1**	**W2**	**W3**	**W4**	**W5**	**W6**	**W7**	**W8**	**W9**	**W10**	**W11**
*S. aureus*	20	>30	30	20	20	10	20	8	20	10	10	10
*E. coli*	30	30	>30	>30	20	20	15	14	20	20	20	20

**Table 2 membranes-12-01180-t002:** Percentage of polyproline II secondary structure calculated according to Equation (2) for the peptides with best MBC presented in Table 1. Values are shaded in blue from light to dark for low-to-high values.

	**2 mM PBS**									**30% TFE**								
**Class**	**R11**	**A3**	**A10**	**P1**	**P6**	**P7**	**L3**	**L9**	**W7**	**R11**	**A3**	**A10**	**P1**	**P6**	**P7**	**L3**	**L9**	**W7**
**PPII** _(Equation (2))_	56	49	47	48	57	44	73	54	59	46	47	46	35	33	31	58	55	51
	***E. coli* membranes**									**DMPG**								
**Class**	**R11**	**A3**	**A10**	**P1**	**P6**	**P7**	**L3**	**L9**	**W7**	**R11**	**A3**	**A10**	**P1**	**P6**	**P7**	**L3**	**L9**	**W7**
**PPII** _(Equation (2))_	52	51	58	54	43	44	71	54	46	71	45	57	48	36	42	73	55	35

## Data Availability

Not applicable.

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
