# Peer review of "Arginine Homopeptide of 11 Residues as a Model of Cell-Penetrating Peptides in the Interaction with Bacterial Membranes"

_membranes, 2022, doi:10.3390/membranes12121180_

Round 1

Reviewer 1 Report

In the paper “R11 homopeptide as a model of cell penetrating peptides in the interaction with bacterial membranes” the Authors use R11, a model polyArg cell penetrating peptide (CPP), to investigate the effect of systematic substitution of the Arg residue in each position by Ala, Pro, Leu or Trp. In this way, they synthesize 44 peptides by the scanning technique, in addition to the model R11.

For each peptide, the antibacterial activity is determined against S. aureus and E. coli, the membrane permeabilization activity is measured by the SYTOX green assay, and their structure is analyzed by CD in four media: PBS, TFE, DMPG LUVs and E. coli membrane extract LUVs.

The paper is of interest to researchers in the fields of CPPs and of antimicrobial peptides, although the results are difficult to rationalize. 

As the Authors comment in the discussion, their work, aimed to shed light on the interaction of model CCPs with bacterial membranes, requires further experiments to be carried out to establish in more detail the effect of each type of substitution in the membrane translocation process. 

Minor revisions

Line 111: 1X107 CFU/ml not 1X107 UFC/ml;

Line 136: what does it mean PBS 2 mM? That the phosphate buffer was 2 mM?

Line 221: Table 2 instead of Table 6;

Line 235: 3.4.1 instead of 3.3.1;

Line 241: Figure 3 instead of Figure 1;

Line 243: Figure 3 instead of Figure 1;

Line 245: 3.4.2 instead of 3.3.2;

Line 252: Figure 4 instead of Figure 2; 

Line 254: 3.4.3 instead of 3.3.3;

Line 262: Figure 5 instead of Figure 3;

Line 264: 3.4.4 instead of 3.3.4;

Line 274: Figure 6 instead of Figure 4.

Editor

This work is clearly written and some of the peptides in which a residue has been replaced do show that antibacterial activity is improved with respect to the starting model peptide. The reasons for improvement are however difficult to understand although they are a good starting point to carry out further experiments with other heteropeptides to understand in more detail the requirements for the membrane translocation process.

Author Response

REVIEWER 1

Comments and Suggestions for Authors

In the paper “R11 homopeptide as a model of cell penetrating peptides in the interaction with bacterial membranes” the Authors use R11, a model polyArg cell penetrating peptide (CPP), to investigate the effect of systematic substitution of the Arg residue in each position by Ala, Pro, Leu or Trp. In this way, they synthesize 44 peptides by the scanning technique, in addition to the model R11.

For each peptide, the antibacterial activity is determined against S. aureus and E. coli, the membrane permeabilization activity is measured by the SYTOX green assay, and their structure is analyzed by CD in four media: PBS, TFE, DMPG LUVs and E. coli membrane extract LUVs.

The paper is of interest to researchers in the fields of CPPs and of antimicrobial peptides, although the results are difficult to rationalize. 

As the Authors comment in the discussion, their work, aimed to shed light on the interaction of model CCPs with bacterial membranes, requires further experiments to be carried out to establish in more detail the effect of each type of substitution in the membrane translocation process. 

We agree with the reviewer on the need to carry out other experiments in the future; however, we believe that with the corrections suggested by all the reviewers and the changes made, the results obtained and their projection are clearer.

Minor revisions

Line 111: 1X107 CFU/ml not 1X107 UFC/ml;

R/ Corrected.

Line 136: what does it mean PBS 2 mM? That the phosphate buffer was 2 mM?

R/ Yes, we used the buffer tablets distributed by Sigma Aldrich (https://www.sigmaaldrich.com/CL/es/product/sigma/p4417). We clarify this in the text.

Line 221: Table 2 instead of Table 6;

Line 235: 3.4.1 instead of 3.3.1;

Line 241: Figure 3 instead of Figure 1;

Line 243: Figure 3 instead of Figure 1;

Line 245: 3.4.2 instead of 3.3.2;

Line 252: Figure 4 instead of Figure 2; 

Line 254: 3.4.3 instead of 3.3.3;

Line 262: Figure 5 instead of Figure 3;

Line 264: 3.4.4 instead of 3.3.4;

Line 274: Figure 6 instead of Figure 4.

R/Thanks for the call of attention; all the numbering was corrected.

Editor

This work is clearly written and some of the peptides in which a residue has been replaced do show that antibacterial activity is improved with respect to the starting model peptide. The reasons for improvement are however difficult to understand although they are a good starting point to carry out further experiments with other heteropeptides to understand in more detail the requirements for the membrane translocation process.

We believe that the revised version of the manuscript, thanks to the reviewers' comments, is now clearer.

Reviewer 2 Report

The manuscript is devoted to the study of the model penetrating peptide R11 homopeptide on the bacterial membranes. I believe that this manuscript is interesting and should be publishable in this journal; however there are several scientific aspects of this manuscript that I feel the authors must first address.

1. line 44-46 The introduction section does not provide complete information about the effect of CCP on membrane. Why the authors do not consider the mechanism of the peptide action due to the formation of the transmembrane pores in the target membranes? This is a well-known mechanism for the antimicrobial peptide. Please, clarify this aspect.

2. line 68 and 94: “2-dimyristoyl-sn-glycero-3-phospoglycerol (DMPG)” should be changed for “1, 2-dimyristoyl-snglycero-3-phosphoglycerol lipid (DMPG)”.

3. Please, clarify the choice of the negatively charge lipids composition DMPG.

4. line 118: “o2.4” should be changed for “2.4”.

5. The measurement temperature of different methods should be indicated somewhere.

6. line 161. “S. aureus” and “E. coli” should be changed for “S. aureus” and “E. coli”.

7. Figure 1 The inscriptions should be larger. Please, check the other figure legends for detail information.

8. Antibacterial activity against S. aureus and E. coli, for different syntactic peptides was studier, however author do not presents the compared this effect with the popular CPP. Why? The positive control with a well-known peptide (not synthetic) must be presented in the cell assay and CD and the conclusions will be substantial.

9. Please, clarify the choice of the concentrations of synthetic peptides using in the work. In this regard, it is necessary to discuss the detergent effects of compound on the properties of lipid membranes.

10. Please, clarify the toxicity of synthesized peptides.

11. The Conclusions section is missing. Please, make a separate section Conclusions.

Author Response

First of all, we want to thank the reviewers and the editor for their enriching comments and for their dedication in reviewing our manuscript entitled “R11 homopeptide as a model of cell penetrating peptides in the interaction with bacterial membranes by Monica Arostica and collaborators.

We have modified the manuscript following the reviewer’s observations and improve the introduction, figures and tables. All the revisions are marked with track changes in MS Word. Additionally, the detailed answers to each reviewer are presented below.

We have done the suggested corrections; and we believe that the manuscript has been improved both in form and content, hoping that now it will be suitable for publication in Membranes.

Sincerely.

Constanza Cárdenas C.

Associate Researcher

Núcleo Biotecnología Curauma

Pontificia Universidad Católica de Valparaíso

REVIEWER 2

Comments and Suggestions for Authors

The manuscript is devoted to the study of the model penetrating peptide R11 homopeptide on the bacterial membranes. I believe that this manuscript is interesting and should be publishable in this journal; however there are several scientific aspects of this manuscript that I feel the authors must first address.

line 44-46 The introduction section does not provide complete information about the effect of CCP on membrane. Why the authors do not consider the mechanism of the peptide action due to the formation of the transmembrane pores in the target membranes? This is a well-known mechanism for the antimicrobial peptide. Please, clarify this aspect.

R/ The introduction was extended, and information about AMPs and their mechanism of pore formation has been included. In the case of CPPs, and particularly the arginine-rich peptides, this mechanism does not apply, since its target is not the lipid membrane.

  1. line 68 and 94: “2-dimyristoyl-sn-glycero-3-phospoglycerol (DMPG)” should be changed for “1, 2-dimyristoyl-snglycero-3-phosphoglycerol lipid (DMPG)”.

R/ Corrected

  1. Please, clarify the choice of the negatively charge lipids composition DMPG.

R/ This was included in the introduction

  1. line 118: “o2.4” should be changed for “2.4”.

R/ Corrected

  1. The measurement temperature of different methods should be indicated somewhere.

R/ Thanks for noticing; it is now included in the description of each method.

  1. line 161. “ AUREUS” and “E. COLI” should be changed for “S. aureus” and “E. coli”.
  2. Figure 1 The inscriptions should be larger. Please, check the other figure legends for detail information.

R/ Thanks for the remark. The figure was changed, and the analysis was made considering the relation of each peptide with R11 (MBC peptide/MBC R11), as suggested by another reviewer. The legends and information on the graph was enlarged.

  1. Antibacterial activity against  aureus and E. coli, for different syntactic peptides was studier, however author do not presents the compared this effect with the popular CPP. Why? The positive control with a well-known peptide (not synthetic) must be presented in the cell assay and CD and the conclusions will be substantial.

R/ We do not understand what the reviewer means with a popular CPP. The comparison was made with the original R11 peptide, which has been reported in other works. Additionally, in our work we did not include natural peptides, meaning natural ones as those directly extracted from their source, because they are very difficult to obtain. Usually in research work what is done is that when their sequences are determined, the peptides are synthesized for the tests to be carried out.

  1. Please, clarify the choice of the concentrations of synthetic peptidesusing in the work. In this regard, it is necessary to discuss the detergent effects of compound on the properties of lipid membranes.

R/ Concentrations of the peptides for each assay were clarified. Arginine-rich CPPs do not target the bacterial membrane. This has been widely reported in the literature; therefore, the detergent effect, which occurs in the carpet model of the interaction of cationic AMPs is not discussed. In addition, in the green SYTOX assay, according to the technique (Bourbon et al 2008) with the concentration used (200 µM), which is higher than the concentration determined in the antibacterial assay, if the membrane were affected, it would be possible see an increase in fluorescence. According the reference (Bourbon et al 2008):“The SYTOX Green nucleic acid stain, which can penetrate cells with compromised plasma membrane but will not cross the membranes of live cells. Indeed, it was demonstrated that bacterial suspensions labeled with SYTOX Green stain emitted green fluorescence in proportion to the fraction of permeabilized cells in the population”.

  1. Please, clarify the toxicity of synthesized peptides.

R/ We have included an assay of hemolytic activity, with the analogs that exhibited higher antibacterial activity than R11 for the two tested bacteria. Additionally, in previous works it was reported that the parent peptide R11 is not cytotoxic.

  1. The Conclusions section is missing. Please, make a separate section Conclusions.

R/According the author instructions this section is optional. Now in the reviewed version we have included the conclusions section.

Reviewer 3 Report

All the comments on the manuscript are reported in the attached pdf.

Author Response

First of all, we want to thank the reviewers and the editor for their enriching comments and for their dedication in reviewing our manuscript entitled “R11 homopeptide as a model of cell penetrating peptides in the interaction with bacterial membranes by Monica Arostica and collaborators.

We have modified the manuscript following the reviewer’s observations and improve the introduction, figures and tables. All the revisions are marked with track changes in MS Word. Additionally, the detailed answers to each reviewer are presented below.

We have done the suggested corrections; and we believe that the manuscript has been improved both in form and content, hoping that now it will be suitable for publication in Membranes.

Sincerely.

Constanza Cárdenas C.

Associate Researcher

Núcleo Biotecnología Curauma

Pontificia Universidad Católica de Valparaíso

REVIEWER 3

In this work, entitled “R11 homopeptide as a model of cell penetrating peptides in the interaction with bacterial membranes” by Arostica M. et al., the authors report about the biolgical acitvity as well as on the conformational behaviour of a homopeptide composed of 11 Arg residues (R11). Then, using the scanning technique, they repleced systematically the Arg with Ala, Pro, Leu and Trp. The obtained peptides were subjected to the same analysis as R11.

In general, the manuscript is well written, even if it can be improved. The authors accomplished the synthesis of 45 peptides, coupled with biological assays and CD analysis. Thus, a huge amount of work was performed. I found the manuscript very interesting and I think that I can attract the attention of readers. Thus, I can recommend publication on Membranes. However, some changes are needed.

  • Please improve the English style, there are some Please, ready it carefully;

R/ Thanks for the comment; the manuscript was reviewed for the language.

  • In the abstract it is written (line 33) that “changes in structure stability were observed”. This sentence is not clear. Stability respect to what? Please, change;

R/ Thanks for the remark; we mean a better definition of the secondary structure, this was corrected in the manuscript

  • The introduction is very short. I think that more information about CPPs (and AMPs) should be added for the benefit of readers, expecially the not expert ones (see 10.3390/ijms22062857). Also some examples can be reported, as the peptide LL-III that is not enriched of Arg but it can act as CPPs (see 3390/ijms22062857);

R/ Thanks for the suggestion, the introduction was extended, and information about AMPs and their mechanism of action has been included. The examples of other CPPs were not included since we want to focus in arginine-rich peptides.

  • For curiosity: is it known the composition of E. coli extracts? If yes, it could be good to add (even if not precise);

R/ A brief description in now included

  • I’m not familiar with the scanning technique. Can the authors add more details about? In addition, why these 4 residues were choosen? Please provide a reason;

R/ In the scanning technique, each peptide residue is replaced by the chosen amino acid. The most widely used amino acid in this technique is alanine, to simulate the "loss" of the side chain, and thus study the specific effect of an amino acid in a defined position. The choice of the 4 specific residues is included in the introduction.

  • Page 3, line 120: a reference is missing;

R/Sorry for the lapse; now reference is indicated.

  • CD section: at which concentrations of peptides and DMPG and E. coli extracts were performed the experiments? I guess that the authors used 2 mM PBS instead of the classic 10 mM (1x) to avoid distortion caused by the high amount of If this is the case, please specify;

R/The reviewer is right. A brief description of each medium is now included in the introduction, and its preparation in the methods section. In particular for PBS we used tablets from Sigma Aldrich. (https://www.sigmaaldrich.com/CL/es/product/sigma/p4417). We clarify this in the text.

  • Page 4, line 151: I agree that HPLC can show about the hydrophilicity, but it does not tell nothing about conformation;

R/ Conformation was eliminated.

  • Figure 1 (pag 5): I guess that the normalization is MBC(R11)/MBC(peptide), so it is not normalized by MBC of R11. This can be a bit confusiong, so I would suggest to report in the opposite way. In this way, the R11 is always 1 and for the more active peptide, the ratio would be less than 1 pointing out a lower MBC;

R/Thanks for bringing it into our attention; the figure was changed.

  • I like the use of Sytox. I can understand that it means that the membrane is not disrupted and, combining the data with biological assays, suggests that the peptides can translocate. However, I recommend to lower down the tone. Unfortunately, it is not a clear proof of translocation (for this, microscopy with GUVs is needed). In addition, have the authors an explanation of why even if the peptides can translocate, they are not able to allow the Sytox to penentrate inside? As pointed out in the discussion, CPPs can be used as carries;

R/ We accept your suggestion and lower the tone. Regarding Sytox and their action on the cells according to the original reference by Bourbon et al 2008 (now included):“The SYTOX Green nucleic acid stain, which can penetrate cells with compromised plasma membrane but will not cross the membranes of live cells. Indeed, it was demonstrated that bacterial suspensions labeled with SYTOX Green stain emitted green fluorescence in proportion to the fraction of permeabilized cells in the population”.

  • Page 7, this is Table 2, no 6;

R/ Corrected

  • All the CD figures: it is really difficult to see the spectra. Numbers are really small and also the legend. Could the authors try to improve them? In addition, at page 8, this is Figure 3 and so on;

R/ The figures were improved and the numbering was corrected. Additionally, we sent the original files of the figures, because the quality is affected in the construction of the pdf file for review.

  • As I can understand, the conclusion of CD is that all the peptides (more or less) can adopt PPII structure, even in neat buffer. So, the PPII is not essential for their translocation? Or in other words: is there a relation between the activity and the conformation?

R/ With the experiments carried out, it was not possible to find a correlation of the activity with the secondary structure, since all the peptides conserved the PPII. Arguably not essential for its antibacterial activity, however it should be seen if an analogue with a non-PPII structure also has CPP activity.

  • The deconvolution with the software gave results in complete disagreement with the use of equation 1. This sounds really strange. But I can understand that sometimes software can fail. The authors have an explanation for that? In addition, I would suggest to report in the main text only the results of equation The results of the software in the Supplementary.

R/ We transferred the data obtained with deconvolution to the supplementary material. One comment in this regard is that there is no reliable software for this type of analysis with peptides, and in particular with secondary structure of PPII. Normally this is predicted as a random coil or unordered; for this reason it is important to take into account the shape of the curve.

Round 2

Reviewer 2 Report

Accept in present form

Author Response

Thanks the text was reviewed.